REGISTERED REPORT PROTOCOL

# Is it easier to use one language variety at a time, or mix them? An investigation of voluntary language switching with bidialectals

**Mathieu Declerck[1], Neil W. Kirk****[2]***

**1** Vrije Universiteit Brussel, Brussels, Belgium, **2** Abertay University, Dundee, Scotland, United Kingdom

* n.kirk@abertay.ac.uk

## Abstract

Previous language production research with bidialectals has provided evidence for similar language control processes as during bilingual language production. In the current study, we aim to further investigate this claim by examining bidialectals with a voluntary language switching paradigm. Research with bilinguals performing the voluntary language switching paradigm has consistently shown two effects. First, the cost of switching languages, relative to staying in the same language, is similar across the two languages. The second effect is more uniquely connected to voluntary language switching, namely a benefit when performing in mixed language blocks relative to single language blocks, which has been connected to proactive language control. If a similar pattern could be observed with bidialectals in a voluntary language switching paradigm, then this would provide additional evidence in favor of similar control processes underlying bidialectal and bilingual language production.

## Introduction

Previous research has indicated that when bidialectals (i.e., speakers of a regional dialect that are also fluent in a standard language variety) produce language, both the standard language and dialect are activated (e.g., [1, 2]), which is assumed to lead to competition among both language varieties. Similar to bilinguals (for reviews, see [3, 4]), a language control process is assumed to be implemented to deal with the competition between language varieties in bidialectals [1, 2, 5, 6]. Some studies have suggested that the language control process implemented by bidialectal speakers of closely related language varieties is similar to the bilingual language control process [1, 2, 5]. In the current study, we set out to further investigate the language control process implemented by bidialectals and its relation to the bilingual language control process by letting bidialectals perform in a voluntary language switching paradigm.

Voluntary language switching [7–16] usually requires participants to name pictures in one of two languages. Unlike other variants of the language switching paradigm (for a review, see [4]), which indicate the language that should be used for each stimulus through cues (e.g., differently colored frames around the stimuli [17]), alternating languages (e.g., AABBAABB, with

**Data Availability Statement:** All relevant data from this study will be made available upon study completion.

**Funding:** This research is funded by a Carnegie Trust for the Universities of Scotland Research Incentive Grant (https://www.carnegie-trust.org/) awarded to NWK (RIG009864). The funders had and will not have a role in study design, data collection and analysis, decision to publish, or preparation of the manuscript.

**Competing interests:** The authors have declared that no competing interests exist.

A and B referring to trials in different languages [18]), or written words (e.g [19]), voluntary language switching allows the participants to choose the language on each trial.

Similar to other variants of language switching (e.g., [17, 20–23]), voluntary language switching with bilinguals generally results in a cost when switching languages, relative to staying in the same language across trials [8–16] however, see [7]. Indicative of the voluntary language switching paradigm with bilinguals is that these switch costs are similar across languages (i.e., symmetrical switch costs [8–14]). This is in contrast with other variants of the language switching paradigm, where asymmetrical switch costs, which entails larger switch costs in the first language (L1) than in the second language (L2) and is typically used as a measure of inhibitory control (e.g., [17, 24–26]for a review, see [27]), are relatively often found. According to Gollan and Ferreira [11], the absence of asymmetrical switch costs with the voluntary language switching paradigm is because this paradigm allows bilinguals to name "easier" words in their L2. This should result in a more similar L1 and L2 activation level for the produced words regardless of language proficiency, and thus might lead to symmetrical switch costs.

Another measure of language control are mixing costs (for a review, see [28]). Mixing costs entail worse performance in repetition trials in mixed language blocks relative to performance in single language blocks. This measure has been explained with control processes that are implemented in anticipation of any upcoming cross-language competition (i.e., proactive language control; e.g., [26]) and the mental cost to maintain and monitor two languages (e.g., [8]). Mixing costs are a highly stable effect in all variants of language switching (e.g., [20, 25, 26, 29–31]), with the exception of voluntary language switching. Voluntary language switching studies actually tend to show a mixing benefit (i.e., worse performance in single language blocks relative to performance in repetition trials in mixed language blocks) in one [11, 12] or both languages [8, 9, 13, 14]. The mixing benefit in L2 observed by Gollan and Ferreira [11] was explained by assuming that only "easier" words are produced in L2 in the voluntary language switching paradigm. Based on this explanation, one would expect that the mixing benefit was only observed for words that were consistently named in L2, but that was not the case [11]. An alternative explanation that can account for a mixing benefit across both languages comes from de Bruin et al. [8]: Production in a single language block requires substantial proactive inhibition of the non-target language [32, 33]. When producing in a mixed language block with voluntary language switching, bilinguals do not require substantial proactive control processes to guide language production, since participants can choose which language to use on any given trial. So, because the implementation of more proactive control processes during single than mixed language blocks is more taxing, better performance is expected in the latter block type when using a voluntary language switching paradigm. From this overview, it appears that voluntary vs. involuntary language switching has a large impact on measures of the bilingual language control process.

In the current study, we set out to investigate if this is also the case for bidialectals by letting bidialectals perform in a voluntary language switching paradigm. The few studies that investigated control processes with bidialectals and bilinguals provide evidence for a shared language control process. Similar to bilinguals, bidialectals show a cost to switching between language varieties, relative to staying in the same language variety across languages [1, 2, 5, 6] Kirk and colleagues [5], for instance, let bidialectals (English-Orcadian) name pictures during a language switching task, where the language variety on each trail was indicated by differently colored frames corresponding to each language variety (cf. involuntary language switching). The results showed that switching between language varieties results in worse performance than staying in the same language variety across trials. Similar to bilinguals (e.g., [17]), these bidialectals showed asymmetrical switch costs, with larger switch costs in their more dominant language variety (dialect) than in their less dominant language variety (standard language; see

also [2]). Asymmetrical switch costs were even found with new bidialectals (English-Dundonian), whereas more fluent bidialectals showed symmetrical switch costs [1]. The latter pattern is similar to that observed with second language learners and highly proficient bilinguals, respectively ([21, 34]). Finally, Kirk and colleagues also showed worse performance in repetition trials in mixed language blocks than in single language blocks [5]. So, along the lines of prior bilingual studies (e.g., [26, 29–31]), mixing costs can be observed with bidialectals during involuntary language switching.

The similarities of bidialectals and bilinguals in previous studies that relied on involuntary language switching seem to indicate that similar language control processes are implemented by these two groups during language production. While it might seem obvious that bilinguals and bidialectals rely on the same language control processes, previous related research indicates that is not necessarily the case. For instance, control processes used within the same language has shown to be different to control used between languages [35]. Even more damning for the assumption that similar control processes are used throughout language processing is that some studies found evidence that different language pairs do not necessarily converge when it comes to language control [36]. A similar discrepancy in language control has been observed across modalities within the same bilinguals (e.g., [37]). These studies provide evidence against the claim that language control is domain general [17, 24], as the control processes within a domain (i.e., language processing) are sometimes even different.

There have also been attempts to objectively distinguish languages from dialects (and thus bilinguals from bidialectals) on a cognitive level using the picture word interference paradigm, which initially suggested dialect items were processed as within-language competitors, akin to synonyms [38]. However, more recent evidence has challenged some of these findings [39, 40]. Thus, the extent to which bidialectals are similar to bilinguals is still unclear and can have theoretical and methodological implications for research comparing bilinguals and monolinguals. For example, research suggesting that there is a general executive control advantage for bilinguals over monolinguals as a result of the regular engagement of language control mechanisms [41], could be invalidated by the presence of bidialectal speakers who also use these mechanisms, but who are erroneously categorized as monolingual [42].

To further investigate the issue of whether there are similar language control processes used in bilingual and bidialectal language production, we set out to examine whether a similar pattern can be observed with bidialectals as with bilinguals in a voluntary language switching paradigm. In the current study, we rely on Scottish speakers of a specific type of Scots as the dialect of interest. Although Scots is recognized by the European Charter for Regional or Minority Languages as a separate minority language (from English), it is generally not given this status, with many facing ridicule for suggesting that Scots and English are separate languages [43]. Even a majority of speakers themselves do not hold this view, with one Scottish Government [44] report demonstrating that 65% of respondents consider their use of Scots as "just a way of speaking". Consequently, these speakers are likely to identify as monolingual rather than bilingual—or even bidialectal—especially if language background measures are not sensitive to the existence of non-standard varieties (see, [1, 5]).

More specifically, we will test speakers of Dundonian Scots and (Scottish) Standard English. Dundonian is an urban dialect used in and around the Scottish city of Dundee. Like other urban Scots dialects, it exists as a lower status variety in a diglossic situation with (Scottish) Standard English as the prestige variety [45]. Whereas Dundonian Scots overlaps substantially with its corresponding standard language, there are several notable differences. First, it is characterized by phonetic differences relative to the standard language, such as vowel differences (e.g. Standard English 'pie' / paɪ / vs. Dundonian 'peh' /pɛ/) and monophthongisation (e.g. Standard English 'mouse' /maʊs/ vs. Dundonian 'moose' /muːs/). Second, and most important

for the current study, there are also words entirely different in Dundonian than its corresponding translation equivalent in Standard English (e.g., "crying" in Standard English would be "greetin" in Dundonian).

If bidialectals and bilinguals rely on similar language control processes, we expect to observe symmetrical switch costs in the voluntary language switching paradigm with the English-Dundonian bidialectals, similar to the pattern observed with bilinguals. This finding might simply indicate that bidialectals tend to produce "easier" words in the less proficient language [11]. Observing a mixing benefit with English-Dundonian bidialectals in a voluntary language switching paradigm would be a more unique finding, as this effect has only reliably been observed with bilinguals in a voluntary language switching paradigm. A mixing benefit would indicate that bidialectals implement proactive language control in single language blocks, whereas this is less the case in mixed language blocks [8].

## Method

### Participants

A power analysis on the mixing benefit was run to determine the required number of participants and trials. We chose the mixing benefit, since this effect seems more uniquely connected to voluntary language switching. Along the approach suggested by Brysbaert and Stevens [46], we ran 200 (Monte Carlo) simulations with the simr package [47] on the voluntary language switching data of Jevtović et al. [14], who tested 40 bilinguals with 20 distinct stimuli in 80 trials in single language blocks and 180 trials in voluntary language switching blocks for each participant. The results showed that the setup of Jevtović et al. [14] had a 99.5% chance of showing a mixing benefit. In the current study, we will rely on the same number of stimuli as Jevtović and colleagues. Furthermore, we will rely on a similar number of participants and trials. The number of voluntary language switching trials per participant will be slightly decreased relative to Jevtović et al. (from 180 to 160 trials per participant). That way, we will have a similar number of trials in the voluntary language switching blocks and the single language blocks. To make sure that we have at least the same number of voluntary language switching block trials across participants as Jevtović and colleagues, we increased the number of participants from 40 to 46.

So, 46 active bidialectal speakers of (Scottish) Standard English and Dundonian Scots will be recruited, all between 18 and 60 years old. These bidialectals will receive the same questionnaire as the bidialectals in Kirk et al. [1], which will provide us with a subjective measure of the participant's language proficiency through self-reported proficiency scores, as well as information regarding the interactional context in which each language is used.

The proposed study has received approval from Abertay University's research ethics committee (EMS4259).

### Materials and task

Along the lines of Jevtović et al. [14], 20 pictures will be presented to the bidialectal participants. These pictures correspond to non-cognate names between Standard English (average number of syllables across the English names: 1.55; average Zipf frequency of the English names: 4.24; [48]) and Dundonian (average number of syllables across the Dundonian names: 1.60; Zipf frequency was not calculated for Dundonian names as no database with word frequency exists for the Dundonian dialect. See S1 Appendix for the stimulus list). Each picture will be presented twice, in non-consecutive trials, throughout each block.

## Procedure

The experimental task will be made publicly available as "Open Materials" on the Gorilla platform (http://gorilla.sc).

The picture naming study will be presented online on the Gorilla platform [49]. After providing informed consent, participants will perform a microphone check, in which they name a sentence and then listen to their own recording. A familiarization block will follow the microphone check, in which all 20 pictures will be presented together with the corresponding names in Standard English and Dundonian. Because there is no standardized written form of the Dundonian dialect, participants will have the option to listen to a recording of the Dundonian word spoken by a local speaker. The familiarization phase will be followed by the actual experiment, which will consist of two single language blocks of 40 trials each and four voluntary language switching block of 40 trials each. There are several ways to present the order of single language blocks and mixed language blocks to obtain mixing costs or a mixing benefit [28]. We opt for the setup used in the bilingual voluntary language switching studies of de Bruin et al. [8, 9] and Jevtović et al. [14]: Participants will first see one single language block, followed by the four voluntary language switching blocks, and then again one single language block in the other language variety than the first single language block. The language variety of the single language blocks will be counterbalanced across participants.

Prior to each of the three block types (i.e., English language block, Dundonian language block, and the voluntary language switching blocks), instructions will be displayed pertinent for that block type, emphasizing speed and accuracy to name each picture. Moreover, in the single language blocks, the bidialectals will be instructed to name each picture in the corresponding language throughout the block. Prior to the voluntary language switching blocks the following sentences will be presented (for similar instructions, see [8, 14]): "In the following section, you can name the pictures in either Standard English or Dundonian. You are free to choose which language variety to use for each picture. However, please do not use the same language variety throughout the whole task.". The instructions will be followed by a short demonstration of the task before completing a short practice block consisting of 8 trials. Finally, the participant will perform the experimental block(s).

Each trial will start with a fixation cross in the middle of the screen. After 250 ms, the fixation cross will be replaced by the stimulus for a maximum of 3000 ms. Each trial will end with a brief 50 ms blank screen before the onset of the next trial.

## Data analyses

The raw data and analyses scripts will be available on the Open Science Framework.

The first trial in each voluntary language switching block will be excluded from both the error analyses and reaction time (RT) analyses, since this is neither a switch nor repetition trial. Furthermore, error trials and trials immediately following an error will be excluded from the RT analyses. Trials with RTs faster than 150 ms, slower than 3000 ms, or three standard deviations above participant mean will also be removed.

Similar to previous bilingual voluntary language switching studies (e.g., [9, 10]), we will provide an overview of the mean switch rate of the participants in all conditions. Furthermore, two analyses will be conducted on the RT and error data. The switch analyses will be conducted on the data of the voluntary language switching blocks and will consist of Trial type (switch vs. repetition trials) and Language variety (Standard English vs. Dundonian). The mix analyses will be conducted on the single language blocks and the repetition trials in the voluntary language switching blocks. The latter analyses will consist of Block type (repetition trials

from voluntary language switching blocks vs. trials from single language block) and Language variety (Standard English vs. Dundonian).

The RTs will be analyzed using linear mixed-effects regression modeling [50]. Furthermore, the error data will be analyzed, if we observe enough errors for meaningful analysis (> 5%), using logistic mixed-effects regression modeling [51]. Both participants and items will be considered random factors with all fixed effects and their interactions varying by all random factors [52]. Yet, convergence issues will be taken into account—we will rely on the buildmer package [53] to simplify the model where such issues are encountered. For all two-level factors we will be using effect coding (i.e., -0.5 and 0.5). Finally, $t$- and $z$-values larger or equal to 1.96 were deemed significant [54].

## Supporting information

**S1 Appendix. Proposed standard English and Dundonian non-cognate stimuli.** (DOCX)

## Author Contributions

**Conceptualization:** Mathieu Declerck, Neil W. Kirk.

**Funding acquisition:** Neil W. Kirk.

**Investigation:** Mathieu Declerck, Neil W. Kirk.

**Methodology:** Mathieu Declerck.

**Project administration:** Neil W. Kirk.

**Writing – original draft:** Mathieu Declerck.

**Writing – review & editing:** Mathieu Declerck, Neil W. Kirk.

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
