## [Decision Letter · Decision Letter 0]

10 Jun 2021

PONE-D-21-10597

Is it easier to use one language variety at a time, or mix them? An investigation of voluntary language switching with bidialectals.

Dear Dr. Kirk,

Thank you for submitting your manuscript to PLOS ONE. After careful consideration, we feel that it has merit but does not fully meet PLOS ONE’s publication criteria as it currently stands.

As you will see below, the reviewers believe that your work will make an important contribution to empirical literature. However, they raised some important concerns, especially with regard to its theoretical impact and methodology (e.g., lack of a power analysis).

Thus, we invite you to submit a revised version of the manuscript that addresses the points raised during the review process. Please submit your revised manuscript by August 6th, 2021. If you will need more time than this to complete your revisions, please reply to this message or contact the journal office at plosone@plos.org. Please include the following items when submitting your revised manuscript:

We look forward to receiving your revised manuscript.

Kind regards,

Veronica Whitford, Ph.D.

Academic Editor

PLOS ONE

Journal Requirements:

Reviewers' comments:

Reviewer's Responses to Questions

**Comments to the Author**

1. Does the manuscript provide a valid rationale for the proposed study, with clearly identified and justified research questions?

Reviewer #1: Yes

Reviewer #2: Partly

2. Is the protocol technically sound and planned in a manner that will lead to a meaningful outcome and allow testing the stated hypotheses?

Reviewer #1: Partly

Reviewer #2: Partly

3. Is the methodology feasible and described in sufficient detail to allow the work to be replicable?

Reviewer #1: Yes

Reviewer #2: Yes

4. Have the authors described where all data underlying the findings will be made available when the study is complete?

Reviewer #1: Yes

Reviewer #2: Yes

5. Is the manuscript presented in an intelligible fashion and written in standard English?

Reviewer #1: Yes

Reviewer #2: Yes

6. Review Comments to the Author

You may also provide optional suggestions and comments to authors that they might find helpful in planning their study.

Reviewer #1: I state that the protocol is "partly" technically sound and planned. I have asked that the authors include a few things that would be important for both replicability and in adding to their research. Most importantly, the authors should justify the block order chosen with respect to the single- and mixed-language blocks. Second, the authors should specify how they will determine significance in their mixed-effects models. Third, I suggest that the authors include additional covariates (mean switch rate, self-rated measures from the questionnaire) as covariates in their analyses. Last, I recommend that the authors include additional non-linguistic tasks of switching and proactive/reactive cognitive control, at the very least to understand the participants better, but ideally to also include as covariates in their analyses, particularly because proactive control plays a crucial role in their hypotheses regarding the mixing benefit. Detailed comments are provided in the attached document.

Reviewer #2: This registered report aims to test if there is a switch cost and mixing benefit in a voluntary language switch paradigm in bidialectals. This topic is potentially interesting but I have some concerns which I hope the authors could address.

--First of all it is not entirely clear what is a theoretical motivation to test bidialectals in a voluntary language switch paradigm. The authors reviewed the literature on bilinguals in involuntary and voluntary language switching paradigms, and bidialectals in involuntary switching paradigm. The reason that just no one has done it before does not make it a theoretical interesting topic to look at. Related to the point, what differences would you expect there will be between bidialectals vs bilinguals? And why you think there will be differences? Alternatively, based on the literature, it sounds like bidialectals are by and large similar to bilinguals, so what is the point to test bidialectals then? I’m not saying this topic is not potentially interesting, it is just in general, the theoretical motivation is not entirely clear.

--The authors also indicated in the literature review that whether switch cost is asymmetrical or symmetrical could be related to language proficiency, at least in involuntary switching paradigms. It is not entirely clear how language proficiency could be affecting the switch cost in the voluntary switch paradigms, although the authors predict that they will see symmetrical switch cost. On a related note, there will be only 20 pictures in this experiment, and there will be a familiarization phase. This is essentially boosting proficiency of names of these particular pictures to a very high level, which would not represent the true language proficiency. I hope the authors could consider using more pictures, and reconsider the use of the familiarization block.

--Related to the above point, although the mixed blocks will be in the middle of two single language blocks. I wonder if the authors are concerned with the potential repetition effect. For instance, by the last mixed blocks, participants will have seen the same pictures 8 times, and they will be way too familiar with these pictures, which will significantly decrease their reaction time and error rate.

--That being said, I think the error rate will be too low to show any meaningful results with the current design (i.e., ceiling effect). Additionally, if the authors really want to analyze error rates, they should also separate them out into regular errors (e.g., use a word that does not match the picture), and incorrect language errors for single blocks at least.

7. PLOS authors have the option to publish the peer review history of their article (what does this mean?). If published, this will include your full peer review and any attached files.

Reviewer #1: **Yes: **Dr. Michael A. Johns

Reviewer #2: No

---

## [Author Response · Author response to Decision Letter 0]

30 Jun 2021

Dear Veronica Whitford,

We are sending you the revised version of our manuscript named “Is it easier to use one language variety at a time, or mix them? An investigation of voluntary language switching with bidialectals”. We are grateful for the comments by the reviewers and incorporated them where possible. 

A detailed point-to-point response to the questions of the reviewers is provided below. Where appropriate a reference to the respective changes in the manuscript is given and the major changes are highlighted in the manuscript (word count of manuscript: 3159).

Reviewer 1

1. Most importantly, the authors should justify the block order chosen with respect to the single- and mixed-language blocks.

Reply: As indicated in Declerck (2020), there are five methodological ways in which single- and mixed-language blocks can be presented to participants, all of them have strengths and drawbacks relative to the others. We chose our setup because that is what several previous voluntary language switching studies have used (e.g., de Bruin et al., 2018, 2020; Jevtović et al., 2020). Since we are mainly interested to see if we can observe a similar pattern as bilingual language switching studies, we used a similar methodology as these studies. This information has been added to the revised manuscript (pages 11).

2. Have the authors considered looking at performance over time as a function of trial number? 

In addition, I might also recommend the use of the buildmer (Voeten, 2021) package in R for the automatic simplification and testing of both random and fixed effects.

Reply: We did not consider looking at performance over time as a main analysis, as we are not sure that this would impact our effect of interest (i.e., mixing benefit). Considering that the mixing benefit should be a measure of proactive language control, and thus a top down, anticipatory control process that is assumed to be implemented over an extended period, we do not think that it would substantially change over time, but this is something we may consider as an exploratory analysis.

 Regarding buildmer, we suggested using the same strategy that we used in previous studies for simplifying the model when encountering a convergence issue (see footnote 2 in the original submission). However, we thank you for this very useful suggestion and agree that the buildmer package would be better to simplify the model. Hence, we have changed this in the revised manuscript (footnote 2 on page 13). 

3. How will the significance of fixed effects terms be determined?

Reply: This is a good question, since we forgot to add this to the original submission! Similar to our previous studies (e.g., Declerck, Ivanova, Grainger, & Duñabeitia, 2020; Declerck, Wen, Snell, Meade, & Grainger, 2019), we deem t- and z-values larger or equal to 1.96 significant (Baayen, 2008). This information has now been added to the manuscript (page 13).

4. Will the authors examine any covariates in their analyses, such as including mean switch rate or self-rated measures from the questionnaire?

Reply: We did not plan to do this, as previous bilingual voluntary language switching studies also did not include any additional covariates. We may include covariates as additional exploratory analyses, but unless they help us explain our main question (Is language control similar across bidialectals and bilinguals?), we do not plan to include these analyses at this point in time.

5. Last, I recommend that the authors include additional non-linguistic tasks of switching and proactive/reactive cognitive control, at the very least to understand the participants better, but ideally to also include as covariates in their analyses, particularly because proactive control plays a crucial role in their hypotheses regarding the mixing benefit.

Reply: We agree that this would be interesting but see this as a part of a potential follow up study, beyond the scope of our proposed study. In the current study, our objective is to establish whether a similar pattern can be observed with bidialectals when voluntary language switching as with bilinguals. To this end, we want to see if we can observe a similar pattern with bidialectals during voluntary language switching.

 Including non-linguistic tasks and comparing those results to our bidialectal switching results would not immediately address our main question, since we know from the bilingual language control literature that studies about shared bilingual and non-linguistic control processes do not always provide a straightforward answer (e.g., Branzi, Calabria, Boscarino, & Costa, 2016; Calabria, Hernandez, Branzi, & Costa, 2012; Declerck et al., 2021). Yet, we think this is an interesting line of enquiry and is definitely an avenue that we will consider for our future research. 

Reviewer 2

1. First of all it is not entirely clear what is a theoretical motivation to test bidialectals in a voluntary language switch paradigm. The authors reviewed the literature on bilinguals in involuntary and voluntary language switching paradigms, and bidialectals in involuntary switching paradigm. The reason that just no one has done it before does not make it a theoretical interesting topic to look at. Related to the point, what differences would you expect there will be between bidialectals vs bilinguals? And why you think there will be differences? Alternatively, based on the literature, it sounds like bidialectals are by and large similar to bilinguals, so what is the point to test bidialectals then? I’m not saying this topic is not potentially interesting, it is just in general, the theoretical motivation is not entirely clear.

Reply: It is not a given that bidialectals and bilinguals rely on similar control processes. Several studies have provided evidence against the notion that control processes are shared during language processing. In turn, this provides evidence against the theoretical assumption that language control is domain general (e.g., Green, 1998; Meuter & Allport, 1999). We have made this motivation clearer now in the introduction (page 6).

 It is true that the few studies that investigated bidialectals found similar effects as bilinguals (e.g., asymmetrical switch costs and mixing costs). However, it should be noted that only a handful of studies have looked into this issue, making additional research a necessity (one would not draw firm conclusions from about 5 studies). Second, the effects that have been observed so far with both bidialectals and bilinguals, and thus have been taken as evidence for an overlap, have also been found in a myriad of non-bilingual and even non-linguistic contexts. The mixing benefit in voluntary language switching seems to be unique to bilingual voluntary language switching. Hence, we think that investigating this effect will provide a strong indication whether bidialectals rely on similar control processes as bilinguals. 

To further strengthen the motivation for this study, we have also included a new section in the introduction outlining some of the implications that can arise as a result of not properly accounting for bidialectal language experience, namely, that some speakers might be erroneously categorized as monolingual because sociolinguistic factors can influence how much prestige and recognition is given to the dialect (see pages 7-8).

2. The authors also indicated in the literature review that whether switch cost is asymmetrical or symmetrical could be related to language proficiency, at least in involuntary switching paradigms. It is not entirely clear how language proficiency could be affecting the switch cost in the voluntary switch paradigms, although the authors predict that they will see symmetrical switch cost. On a related note, there will be only 20 pictures in this experiment, and there will be a familiarization phase. This is essentially boosting proficiency of names of these particular pictures to a very high level, which would not represent the true language proficiency. I hope the authors could consider using more pictures, and reconsider the use of the familiarization block.

Reply: Language proficiency should not immensely affect the outcome of (a)symmetrical switch costs during voluntary language switching, according to the interpretation that Gollan and Ferreira (2009) gave for their symmetrical switch costs in a voluntary language switching study. We have made this clearer now in the introduction (page 4).

Unfortunately, we have exhausted all possible depictable noncognate words between Dundonian Scots and Standard English. Hence, we will not be able to increase the number of stimuli. Yet, we do not believe that this is a major issue. Plenty of bilingual language switching studies have relied on far less stimuli and still found asymmetrical switch costs (e.g., Meuter & Allport, 1999) and mixing costs (e.g., Stasenko, Matt, & Gollan, 2017) with the cued language switching paradigm. Moreover, using a similar methodology and 20 stimuli, Jevtović and colleagues (2019) found the same pattern that we aim to investigate. A similar argument could be made for the familiarization block (e.g., Li & Gollan, 2021; Peeters & Dijkstra, 2018), with which we have even found asymmetrical switch costs with bidialectals in an involuntary language switching paradigm (Kirk et al., 2018, 2021). Grunden et al. (2020) also used a familiarization block in their bilingual voluntary language switching study, and still observed a mixing benefit. Since previous research indicates that neither 20 stimuli nor a familiarization block are detrimental to observing the effects we are interested in, we would keep both these features of our setup.

3. Related to the above point, although the mixed blocks will be in the middle of two single language blocks. I wonder if the authors are concerned with the potential repetition effect. For instance, by the last mixed blocks, participants will have seen the same pictures 8 times, and they will be way too familiar with these pictures, which will significantly decrease their reaction time and error rate.

Reply: The mixed language blocks will benefit more, relative to the first single language block, and less, relative to the last single language block, from any potential repetition effects. Hence, this repetition effect should be balanced out to a large degree by the block order. Furthermore, studies with far less stimuli (< 10 stimuli) than the proposed current study (20 stimuli) have shown evidence for a mixing cost with cued language switching (e.g., Stasenko et al., 2017) and alternating language switching (e.g., Declerck, Philipp, & Koch, 2013). More to the point for the current study is that several previous bilingual voluntary language switching studies have shown a mixing benefit with the same (Jevtović et al., 2019), or a similar (de Bruin et al., 2018), number of stimuli and a similar setup. Hence, if similar control processes are implemented by bilinguals and bidialectals, then we should observe a similar outcome (i.e., mixing benefit) as those methodologically similar bilingual studies.

4. That being said, I think the error rate will be too low to show any meaningful results with the current design (i.e., ceiling effect). Additionally, if the authors really want to analyze error rates, they should also separate them out into regular errors (e.g., use a word that does not match the picture), and incorrect language errors for single blocks at least.

Reply: We agree that the error rate will probably be too low to show meaningful results. For this reason, we included the following sentence in the original submission: “Furthermore, the error data will be analyzed, if we observe enough errors for meaningful analysis (> 5%), using logistic mixed-effects regression modeling (Jaeger, 2008).” (page 12).

 We also agree that it makes sense to differentiate between lexical errors and language errors in cued language switching studies. However, we are not sure that this also applies to voluntary language switching studies, as it is unclear whether the participant will have produced in the correct or incorrect language variety in the mixed language block. We acknowledge that the reviewer wants us to use this specific error coding for single language blocks, but it is unclear what we would gain from this, as the single language block performance is compared to the mixed language block performance. Since it does not make sense to compare the error rate of, for example, language errors in single language blocks to the overall error rate in mixed language blocks, we did not include this adjusted error coding schema in the revised manuscript.

Additional changes

1. We have added more information about Scots, as this will provide more sociolinguistic context for the reader regarding the status of this language variety in Scotland (page 8).

We would like to thank you and the reviewers for these helpful comments. We hope you find our revision and response to the reviewers’ comments satisfactory.

Sincerely,

Mathieu Declerck and Neil W. Kirk

---

## [Decision Letter · Decision Letter 1]

10 Aug 2021

Is it easier to use one language variety at a time, or mix them? An investigation of voluntary language switching with bidialectals.

PONE-D-21-10597R1

Dear Dr. Kirk,

We’re pleased to inform you that your manuscript has been judged scientifically suitable for publication and will be formally accepted for publication once it meets all outstanding technical requirements.

Kind regards,

Veronica Whitford, Ph.D.

Academic Editor

PLOS ONE

Additional Editor Comments (optional):

Reviewers' comments:

Reviewer's Responses to Questions

**Comments to the Author**

1. Does the manuscript provide a valid rationale for the proposed study, with clearly identified and justified research questions?

Reviewer #1: Yes

Reviewer #2: Yes

2. Is the protocol technically sound and planned in a manner that will lead to a meaningful outcome and allow testing the stated hypotheses?

Reviewer #1: Yes

Reviewer #2: Yes

3. Is the methodology feasible and described in sufficient detail to allow the work to be replicable?

Reviewer #1: Yes

Reviewer #2: Yes

4. Have the authors described where all data underlying the findings will be made available when the study is complete?

Reviewer #1: Yes

Reviewer #2: Yes

5. Is the manuscript presented in an intelligible fashion and written in standard English?

Reviewer #1: Yes

Reviewer #2: Yes

6. Review Comments to the Author

You may also provide optional suggestions and comments to authors that they might find helpful in planning their study.

Reviewer #1: I would like to thank the authors for addressing my comments in this revision. With regards to my main comments:

1) The justification for the chosen block order is understandable, thank you for clarifying. It is indeed a good decision to match the methodology to previous language-switching studies, so I have no further issues with this.

2) You would be surprised what processes can change throughout the course of a single task! Though I completely understand this line of thought, that proactive control should be more-or-less executed at the task level. If you do end up looking at this, even just a graph, and find something interesting, do let me know.

3) This is perhaps my only remaining issue, but one that is very easily addressed. While it is true that a t- or z- value greater than ~1.96 suggests significance, I believe it is also important to do some further model testing. Specifically, the authors could perform model comparisons using the anova() function in R to compare a model with the effect of interest to a model without the effect of interest, with all other variables staying the same. The anova() function performs a chi-squared test on the residual variance and indicated whether including the effect of interest significantly improves model fit. Other ways to do this are using the AIC or BIC values (smaller values are better fitting models), using the Anova() (capital A!) function in the car library in R to perform an omnibus test, or--the most convenient--using buildmer, which will automatically perform testing of each effect via backwards elimination. I recommend using Satterthwaite ddf, which is an argument that can be changed in the buildmer function (the default is Wald, which is also acceptable). Whatever paths the authors may choose, I ask that they simply perform so further model criticisms aside from just relying on the t- or z- values from the model. Pairing this with another method described above would be excellent, as well.

4) Again, this is understandable, that you would want to keep the methodology similar to previous language-switching studies. Do keep in mind these variables for post-hoc and follow-up analyses, however, as individual differences can go a long way in soaking up variance that may inadvertently make it into the fixed effects.

5) Again, this is understandable to match prior studies. As you mention, however, I hope that you do keep it in mind in your future research, if only to satisfy the curiosity of readers (and reviewers).

Reviewer #2: The authors have addressed my previous concerns. The study design is sound and the topic is interesting. I don't have any further comments.

7. PLOS authors have the option to publish the peer review history of their article (what does this mean?). If published, this will include your full peer review and any attached files.

Reviewer #1: No

Reviewer #2: No

---

## [Editor Report · Acceptance letter]

20 Aug 2021

PONE-D-21-10597R1 

Is it easier to use one language variety at a time, or mix them? An investigation of voluntary language switching with bidialectals. 

Dear Dr. Kirk:

I'm pleased to inform you that your manuscript has been deemed suitable for publication in PLOS ONE. Congratulations! Your manuscript is now with our production department. 

Kind regards, 

on behalf of

Dr. Veronica Whitford 

Academic Editor

PLOS ONE